# Sulphated TiO_2_ Reduced by Ammonia and Hydrogen as an Excellent Photocatalyst for Bacteria Inactivation

**DOI:** 10.3390/ma17010066

**Published:** 2023-12-22

**Authors:** Piotr Rychtowski, Oliwia Paszkiewicz, Agata Markowska-Szczupak, Grzegorz Leniec, Beata Tryba

**Affiliations:** 1Department of Catalytic and Sorbent Materials Engineering, Faculty of Chemical Technology and Engineering, West Pomeranian University of Technology in Szczecin, Pułaskiego 10, 70-322 Szczecin, Poland; beata.tryba@zut.edu.pl; 2Department of Chemical and Process Engineering, West Pomeranian University of Technology in Szczecin, Piastów 42, 71-065 Szczecin, Poland; oliwia.paszkiewicz@zut.edu.pl (O.P.); agata.markowska@zut.edu.pl (A.M.-S.); 3Department of Nanomaterials Physicochemistry, Faculty of Chemical Technology and Engineering, West Pomeranian University of Technology in Szczecin, Piastów 42, 71-065 Szczecin, Poland; grzegorz.leniec@zut.edu.pl

**Keywords:** photocatalysis, solar light, reduced TiO_2_, *Escherichia coli*, *Staphylococcus epidermidis*, bacterial inactivation

## Abstract

This study presents a relatively low-cost method for modifying TiO_2_-based materials for photocatalytic bacterial inactivation. The photocatalytic inactivation of Gram-negative (*Escherichia coli*) and Gram-positive (*Staphylococcus epidermidis*) bacteria using modified sulphated TiO_2_ was studied. The modification focused on the reduction of TiO_2_ by ammonia agents and hydrogen at 400–450 °C. The results showed a high impact of sulphate species on the inactivation of *E. coli*. The presence of these species generated acid sites on TiO_2_, which shifted the pH of the reacted titania slurry solution to lower values, around 4.6. At such a low pH, TiO_2_ was positively charged. The ammonia solution caused the removal of sulphate species from TiO_2_. On the other hand, hydrogen and ammonia molecules accelerated the removal of sulphur species from TiO_2_, as did heating it to 450 °C. Total inactivation of *E. coli* was obtained within 30 min of simulated solar light irradiation on TiO_2_ heat-treated at 400 °C in an atmosphere of Ar or NH_3_. The *S. epidermidis* strain was more resistant to photocatalytic oxidation. The contact of these bacteria with the active titania surface is important, but a higher oxidation force is necessary to destroy their cell membrane walls because of their thicker cell wall than *E. coli*. Therefore, the ability of a photocatalyst to produce ROS (reactive oxidative species) will determine its ability to inactivate *S. epidermidis*. An additional advantage of the studies presented is the inactivation of bacteria after a relatively short irradiation time (30 min), which does not often happen with photocatalysts not modified with noble metals. The modification methods presented represent a robust and inexpensive alternative to photocatalytic inactivation of bacteria.

## 1. Introduction

Conventional disinfection methods include chemicals, which can produce toxic byproducts, and ultraviolet light, which requires a source of electricity. Solar water disinfection (SODIS) is a simple and cost-effective method to purify drinking water by using solar energy to inactivate waterborne pathogens (e.g., viruses, bacteria, and parasites) related to diseases including cholera, diarrhoea, polio, typhus, and many others [1].

Among photocatalysts, TiO*_2_*-based materials have gained prominence due to their cost-effectiveness, excellent stability, and high activity. However, the material’s wide band gap has prompted numerous efforts to enhance its light absorption within the visible spectrum [2]. The mechanisms behind bacterial inactivation, particularly with modified titanium dioxide, remain a topic of uncertainty, especially when different parameters, effects, or modification methods are taken into account. The varied photocatalyst TiO_2_ has been successfully used under solar irradiation for destroying many types of pathogens [3,4]. The most likely mechanism of bacterial inactivation is based on the generation of reactive radicals, such as ∙OH, which destroy the bacterial membrane and cause damage to cell organelles [5].

Despite these advancements, there is an ongoing need to further elevate the activity of TiO_2_-based materials. Commonly employed modification methods involve noble metals (Pt, Au, or Ag), transition metals (Fe or Cu), and non-metal dopants (H, S, F, N, or C) [6,7]. An intriguing approach involves creating Ti^3+^ surface defects on TiO_2_, which not only improves visible light utilization but also enhances the separation of electron-hole pairs, thereby promoting the generation of reactive radicals [8,9]. This self-doping of Ti^3+^ emerged as a promising strategy to augment the antimicrobial properties of TiO_2_ within the visible spectrum [9]. Some attempts to utilize hydrogenated TiO_2_ for *Escherichia coli* inactivation under visible light were made [10]. Sulphur-doped TiO_2_ shows promising results towards *E. coli* inactivation, where complete inactivation can be achieved using indoor sources of light [11,12]. The presence of sulphur most likely leads to increased harvesting of light from the visible range [12]. A similar effect of increased visible light absorption was caused by nitrogen doping, resulting in increased antibacterial activity of TiO_2_ [13].

The purpose of this research study was the application of a relatively simple and affordable technology for obtaining TiO_2_ with photocatalytic and bacteriostatic properties. This technology assumes modifications of industrially produced sulphated TiO_2_ with no addition of noble and/or semi-noble metals, making it economically and environmentally attractive. TiO_2_ modification involves thermal treatment in atmospheres of ammonia, hydrogen, or argon to produce TiO_2_ surface defects of various types. The impact of the titania surface defects and the presence of sulphate species on TiO_2_’s bacteriostatic properties are discussed. Gram-negative *E. coli* was chosen as the bacterial model organism, since its characteristics make it the obvious organism to study the disinfection properties of photocatalysts. *E. coli* in water is recognized as an indicator of water quality and as an indicator to monitor the level of pathogens in water or wastewater and to measure the efficiency of disinfection treatment. A second strain of choice we decided to study was the genus *Staphylococcus*. *S. epidermidis* does not produce aggressive virulence determinants, but is a ubiquitous colonizer of human skin, which is the most common source of infection on indwelling medical devices. Recently, *Staphylococcus epidermidis* has become a major public health concern. In addition, some strains of *S. epidermidis* are highly salt tolerant and are commonly found in marine environments.

## 2. Materials and Methods

### 2.1. TiO_2_ Preparation

As a photocatalyst nanoparticles source, amorphous titania was used, which was sourced from a chemical factory (Grupa Azoty “Police” S.A., Police, Poland). The production block diagram for the sulphate method is presented in Figure 1. This is the industrial process oriented towards large-scale production. Obtained after filtration and washing, the titania still contained some residues of TiOSO_4_, as well as small post-synthesis iron species in the form of Fe^2+^, which did not exceed a few ppm. Therefore, the titania slurry, as produced, has a slightly acidic pH.

Obtained after this process, the titania was further processed in our research studies. The schematic diagram of further TiO_2_ modification is presented in Figure 2. A two-step process was undertaken to produce photocatalysts. Initially, a water suspension of raw TiO_2_ underwent heating at 150 °C under a naturally elevated pressure of approximately 7.4 bar for a duration of 1 h. During this phase, either distilled water (pretreated TiO_2_) or an ammonia–water solution (pretreated N-TiO_2_) was introduced into the autoclave to maintain either a pH level of 7 or 10, respectively. Subsequently, the resulting pretreated titania was transferred to a pipe furnace and subjected to heat treatment at temperatures of 400 or 450 °C for 2 h under different gas atmospheres. The utilized gases were argon, hydrogen, or ammonia. The applied gas flow was 20 mL/min, and a heating rate of 10 °C/min was employed during this process.

### 2.2. Zeta Potential and pH

Both zeta potential and pH analyses were conducted utilizing the Malvern PANalytical Zetasizer Nano-ZS instrument (Almelo, The Netherlands). Prior to measurement, each titania suspension was prepared in 0.85% NaCl solution, which matched the *E. coli* environment. In each instance, 20 mg of a given photocatalyst was dispersed within 100 mL of the solution using an ultrasonic bath for a duration of 15 min. The suspension’s pH was determined by employing a pH-responsive electrode previously calibrated using three distinct buffers: acidic, neutral, and basic. Such prepared suspensions were subsequently transferred to an electroconducting measurement cell, where zeta potential measurements were performed.

### 2.3. X-ray Diffraction (XRD)

XRD patterns were recorded using an Empyrean Diffractometer PANanalytical (Almelo, The Netherlands) equipped with a copper lamp (λ = 0.154439 nm). Prior to measurement, the lamp parameters were set to 35 kV and 30 mA. The mean crystallites’ size of anatase and rutile were obtained according to Scherrer’s equation:$$D=\frac{K\times \lambda }{\left(\beta -b\right)\times \cos\theta }$$
where K—shape factor (K = 0.93), λ—Cu lamp wavelength (nm), β—FWHM (rad), and θ—diffraction angle (°). Miller indexes describing a given diffraction peak were determined on the basis of the standard diffraction data of JCPDS: 01-071-1168 for anatase [14] and 01-088-117 for rutile [14]. The standard XRD patterns are presented in Appendix A.

### 2.4. X-ray Fluorescence (XRF)

The percentage amount of sulphur content in the TiO_2_-based photocatalysts was measured with an X-ray fluorescence (EDXRF) spectrometer Epsilon3, Malvern PANanalytical (Almelo, The Netherlands). Prior to the measurement, the internal pattern was used.

### 2.5. FTIR Spectroscopy

Fourier transform infrared spectroscopy (FTIR) was utilized to investigate the surface groups present on the surface of the studied photocatalysts. A Jasco 4200 spectrometer using the reflection technique was used for the spectra measurement in the range of 4000–1000 cm^–1^. Background measurement on pure KBr was carried out as a reference and subtracted from the measurement of the photocatalyst samples.

### 2.6. UV-Vis/DR Spectroscopy

To investigate the effects of different conditions of temperature and atmospheres on the optical properties of the studied samples, UV-Vis/DR spectroscopy measurements were performed. A V-650 spectrometer (Jasco, Tokyo, Japan) was used for this purpose. The studied wavelength range was 200–800 nm. A pure BaSO_4_ block was used as the reference.

### 2.7. EPR Spectroscopy

Electron paramagnetic resonance (EPR) is a highly sensitive technique for investigating small quantities of paramagnetic systems such as metal ions with unpaired electrons or crystal lattice defects. Titanium (Ti) ions can exist in three oxidation states: 2, 3, and 4. At room temperature, the EPR signal of only Ti^3+^ can be recorded due to its spin S = 1/2 (one unpaired electron), whereas Ti^2+^ and Ti^4+^ ions have spins S = 1 and S = 0, respectively. Lattice defects, such as trapped electrons or vacancies in the crystal lattice, can be recorded with the EPR technique if they form paramagnetic centres, even from single electrons or holes.

A conventional EMXplus Bruker X-band EPR-CW spectrometer operating in the frequency range 9.2–9.9 GHz and a microwave power of 2.3 mW was used to record EPR spectra at room temperature. The first derivative of the absorption spectrum was measured as a function of the applied magnetic induction up to 700 mT. The position of the EPR line was calculated from the formula:$$g_{1,2,3}=7,144,773\times f_{rez}(GHz)/B_{rez}(mT)$$
where g is the Zeeman splitting factor. The fitting of the parameters of the spin Hamiltonian (g_x_, g_y_, and g_z_) was obtained using the SIMPOW6 software [15] The integrated EPR signal intensity is defined as the static susceptibility of the spins involved in the resonance and referred to as the area under the absorption curve.

### 2.8. Antibacterial Tests

The antibacterial actions of the tested photocatalysts were tested against Gram-negative *Escherichia coli* K12 ATCC 25992 and Gram-positive *Staphylococcus epidermidis* ATCC 49461. The thawed bacteria were suspended in Nutrient Broth for *E. coli* (BioMaxima Sp. z o.o, Lublin, Poland) or BHI Broth for *S. epidermidis* (BioMaxima Sp. z o.o, Lublin, Poland) and incubated for 24 h at 37 ℃. The bacterial pellet was separated by centrifugation (5000 RPM for 10 min) and diluted with a sterile isotonic solution: (a) 0.85% NaCl (Chempur, Piekary Slaskie, Poland) for *Escherichia coli* and (b) phosphate-buffered saline PBS (Chempur, Piekary Slaskie, Poland) for *Staphylococcus epidermidis*. Then, the bacterial suspensions were adjusted to 0.5 McFarland turbidity (using a McFarland densitometer DEN-1, Biosan, Riga, Latvia). The bacterial suspensions were standardized to an approximate number of bacteria 1.5 × 10^8^ CFU × cm^−3^, and the tested photocatalyst was resuspended. The final photocatalyst concentration was 0.1 g × dm^−3^. The photocatalytic process in the dark and under UV-VIS light (lamp ULTRA-VITALUX 230 V E27/ES, OSRAM 300 W, Munich, Germany) was carried out according to the methodology described in our previous paper [3]. The number of surviving bacteria [CFU × cm^−3^] as a function of time is presented as a survivorship curve.

The fluorescence technique was employed to detect the formation of ∙OH radicals on the surface of TiO_2_. This involved converting terephthalic acid (TA) into 2-hydroxyterephthalic acid (2-HTA) while subjecting it to UV-VIS light irradiation (the same as for antimicrobial tests) in the presence of the TiO_2_-based photocatalysts. The concentration of TA utilized was 5 × 10^−4^ mol∙dm^−3^, with a sample weight of 20 mg and a solution volume of 100 cm^3^. To mimic a bacterial environment, NaCl (0.85%) was used as the reaction solution. The concentration of 2-HTA was measured using a F-2500 Fluorescence Spectrophotometer from Hitachi (Kyoto, Japan).

## 3. Results

The zeta potential and pH measurements are presented in Table 1. It can be seen that most studied samples had a pH close to neutral. The deviation from this trend was characterized by samples T-Ar-400 and T-Ar-450, whose pHs were slightly acidic. This was due to the presence of SO_4_^2–^ groups on the surface of those samples. What is more, this also affected their zeta potential, which turned highly positive, while the rest of the samples had a negative zeta potential value. This effect can affect the adsorption of pollutants on the surface of the photocatalyst and either increase or decrease its photocatalytic activity, depending on the type of pollutant [16].

In Figure 3, the XRD patterns of the TiO_2_-based samples are presented. It can be seen that the main crystal phase of all studied samples was anatase. Additionally, minor reflexes of rutile were visible, which indicated a small share of the rutile phase. The proposed two-step synthesis allowed for obtaining highly crystalline materials. Our previous report [17] showed that the first step led to obtaining around 71% crystalline TiO_2_. Therefore, an additional heat treatment at 400 or 450 °C most likely led to even better crystallinity.

In Table 2, the X-ray fluorescent elemental sulphur content and the crystallites’ size (calculated from Scherrer’s formula) of the studied samples are presented. It can be seen that treating TiO_2_ with ammonia–water (T-400-NH_4_OH and T-450-NH_4_OH) led to the dissolution and flushing out of the sulphur compounds. Since some of the samples tested were heat-treated at the aforementioned temperatures and in an atmosphere of a reducing agent, their sulphur content was therefore much lower than those treated at 400 °C. H_2_S can be produced by treating hydrogen with molten elemental sulphur at about 450 °C [18]. Due to the reaction of small amounts of sulphur compounds with gaseous hydrogen or ammonia, hydrogen sulphide was formed and desorbed from the T-450-NH_3_ and T-450-H_2_ samples. On the other hand, the amount of sulphur remained unchanged in samples T-400-Ar and T-450-Ar due to the presence of a protective argon atmosphere.

In Figure 4, the FTIR spectra of the obtained TiO_2_-based photocatalysts are presented. Regardless of the sample, a wide band at the range of around 3800–2500 and a band at 1620 cm^–1^ were observed due to the presence of physisorbed water and hydroxyl groups coordinated to the TiO_2_ surface [19]. A wide band at 1240 cm^–1^ (Figure 4a,b) can be ascribed to the presence of sulphur groups, which originate from the sulphate method of raw TiO_2_ industrial treatment [3]. Its intensity was therefore significantly lower in cases of samples containing less sulphur (Table 2). In cases of photocatalysts modified with nitrogen-based compounds (T-NH_3_ and T-NH_4_OH), two major bands appeared at around 1437 and 1520 cm^–1^ and can be assigned to NH_4_^+^ ammonium ion formed [20] and NH_2_ groups coordinated to the photocatalyst’s surface [21], respectively. The 1437 cm^–1^ band intensity was the strongest for T-NH_3_-400. Interestingly, both T-H_2_-400 and T-H_2_-450 had a broad band at around 1250 cm^–1^, which originated due to the TiO_2_ hydrogenation and formation of Ti_x_H_y_ [22,23].

In order to track the absorption ability of the studied samples, UV-Vis/DR spectroscopy measurements were performed. On the basis of the presented spectra (Figure 5a,b) it can be observed that most studied photocatalysts demonstrated a similar absorption in the UV-Vis range. However, it is worth noting that samples modified with ammonia (T-NH_3_-400 and T-NH_3_-450) showed some increased absorption in the visible range. Samples heat-treated in a 400 °C and ammonia atmosphere showed increased absorption in the range of around 400–550 nm, whereas the samples heat-treated with 450 °C had increased absorption in the range of 400–650 nm. Enhanced adsorption of visible light by TiO_2_ modified with gaseous NH_3_ is a result of the strong adsorption of ammonia species on the TiO_2_’s surface. In cases of other modifications of TiO_2_, such as hydrogenation and doping with an ammonia solution, the modified species were doped not only on the surface but also at the interstitial positions of the TiO_2_ lattice and at the defect sites of TiO_2_, and therefore, they did not significantly change the optical properties of the TiO_2_ surface. In addition to the UV-Vis spectra measurements presented, the band gaps were determined using the Kubelka–Munk method. The results are shown in Appendix A. Two different values were observed: the first one corresponding to the rutile phase (Eg_1_) was about 3.1 eV, and the second one corresponding to the anatase phase (Eg_2_) was about 3.2–3.3 eV. No significant changes were observed in the values of the band-gap energies as a result of the heat-treatment in different atmospheres (NH_3_, H_2_, or Ar).

In the study of the TiO_2_-based materials compounds, the EPR technique was used to detect crystal lattice defects. The bulk TiO_2_ compound contains Ti^4+^ ions, which do not generate EPR signals. Different synthesis methods and conditions result in the formation of crystal lattice defects in the closest environment of the Ti^4+^ ions, creating paramagnetic centres that can be recorded with the EPR technique. Figure 6 shows the EPR spectra obtained for a TiO_2_-based material heat-treated at 400 °C.

Several EPR lines with different intensities were observed on the EPR spectra, depending on the synthesis conditions. For the photocatalysts prepared in the NH_4_OH atmosphere, we observed only one weakly intense EPR line (g_1_ = 2.0014) originating from electrons trapped in the crystal lattice site (the red curve in Figure 6). Adsorption of an oxygen molecule on reduced TiO_2_ led to the appearance of an O_2_^–^ EPR signal. In the cases of TiO_2_-based photocatalysts prepared in Ar or H_2_ atmospheres and heat-treated at 400 °C, additional EPR lines were observed. An EPR line was observed at g_2_ = 1.967, which originated from electrons at interstitial sites due to the adsorption of oxygen on the reduced rutile form of TiO_2_. The EPR signals were very close, but the positions of the EPR lines were different. The reason for the differences in the positions of the EPR lines may be the interactions of the crystal field, which causes a shift of the O_2_^–^ EPR line towards higher magnetic inductions. The signal at g_3_ = 1.933 originated from Ti^3+^ ions for the TiO_2_ in rutile form (arising from an oxygen defect associated with trivalent titanium ions), and the broad, intense EPR line originated from Ti^3+^ ions in the TiO_2_ in anatase form.

In the cases of the TiO_2_-based photocatalysts produced in the NH_3_ atmosphere, we observed two EPR lines. The first originated from electrons at the interstitial site (g_2_ = 1.967), and the second one was a broad and intense EPR line originating from Ti^3+^ ions in TiO_2_ in anatase form. The anatase form shows rhombic symmetry with spin Hamiltonian parameters g_x_ = 2.001, g_y_ = 1.994, and g_z_ = 1.976 [15]. The parameters of the spin Hamiltonian are the same for all anatase forms of the TiO_2_ compound. This implies that the nearest environment for titanium ions is the same for three of the four atmospheres of TiO_2_ material fabrication. In the fourth sample prepared with the addition of NH_4_OH, we did not observe signals from Ti^3+^ defects. The highest number of Ti^3+^ defects was observed for the anatase form of TiO_2_ produced in an NH_3_ atmosphere, and it was characterized by the absence of the rutile form in the TiO_2_ material (no signal at g_3_) as well as a lack of signal from electrons in the interstitial sites (no signal at g_2_). It is also worth noting that the signal from electrons trapped in the crystal lattice sites (signal at g_1_) was present in all TiO_2_-based photocatalysts, but for three of them, it was in superposition with the considerably more intense signal from the Ti^3+^ defects.

TiO_2_-based photocatalysts were also obtained at 450 °C and their EPR spectra are presented in Figure 7.

The positions of the EPR lines of TiO_2_-based photocatalysts heat-treated at 450 °C were very similar or identical to the positions of the EPR lines obtained for TiO_2_-based photocatalysts heat-treated at 400 °C, but the intensities of the EPR lines were different. For ones obtained at 450 °C, and in NH_3_ and H_2_ atmospheres, signals from the Ti^3+^ defects of the anatase form and Ti^3+^ defects of the rutile form were observed, as well as electrons trapped in the interstitial crystal lattice sites. The intensities of the lines increased significantly as the number of Ti^3+^ defects of the rutile phase increased (see Figure 8). The intensity of the signal from electrons trapped in the interstitial crystal lattice sites also increased. This indicates that the interstitial sites of electrons could be related to the rutile phase of TiO_2_. It is worth noting that for TiO_2_ heat-treated at 450 °C in an H_2_ atmosphere (T-H_2_-450), two broad EPR lines were observed (see Figure 8, bottom panel, arrows). They probably originated from free electrons in the conduction band. As the temperature T > 450 °C increases, the intensity of this signal increases, as we can observe in our previous paper [24].

In Figure 9a,b and Figure 10a,b, the results of the microbial tests conducted in dark conditions are presented. Almost all examined photocatalysts had very poor antibacterial properties against *S. epidermidis* and *E. coli* when the light was off.

The bactericidal activity of the tested photocatalysts was effective only upon solar irradiation. However, the model of action depended on the type of bacteria. Gram-positive bacteria (*S. epidermidis*) were more resistant to the photocatalytic process. After 30 min of irradiation, a significant reduction (>2 log) was observed for the T-NH_4_OH-400, T-H_2_-450, and T-Ar-450 samples (Figure 9c,d). In the same conditions, total inactivation (>5 log) of the Gram-negative *E. coli* bacteria was achieved by T-Ar-400 and T-NH_3_-400 (Figure 10c). A satisfactory outcome of bacterial reduction > 3.5 log) was also obtained for T-H_2_-400, T-H_2_-450, T-NH_3_-450, and T-Ar-450 (Figure 10c,d). The weakest antibacterial property in relation to both tested bacteria was presented by T-NH_4_OH-450 irradiated under solar light (Figure 9d and Figure 10d).

In Figure 11, ∙OH radicals generation efficiency measurements are presented. The tests were performed in either 0.85% NaCl or PBS media, which correspond to the environment of *E. coli* and *S. epidermidis*, respectively, for better reliability of the radical formation effect on the antimicrobial properties of the TiO_2_-based photocatalysts. It can be seen that independent of the process environment, the best performance was observed in the cases of the T-NH_4_OH-400 and T-NH_4_OH-450 photocatalysts. The lowest amount of radicals formed was observed when T-H_2_-450 was used.

## 4. Discussion

The performed studies showed that sulphated titania is a very attractive feedstock for the production of photocatalysts that have antibacterial properties towards *E. coli* inactivation. The proposed mechanism of photocatalytic *E. coli* inactivation is shown in Figure 12. Since sulphur is considered an active element with antibacterial, antifungal, and antiviral properties [25], the sulphur content of the prepared TiO_2_-based photocatalysts is a significant factor in determining their antibacterial potential for *E. coli*. It has been demonstrated by a few research teams that S-doped TiO_2_ exhibited strong antibacterial activity against selected strains [12,26,27]. The photocatalytic activity of the TiO_2_-based samples obtained in the present study with increased sulphur content, such as T-NH_3_-400 (1.19%) and T-Ar-400 (1.40%), had increased antimicrobial activity against *E. coli* under solar irradiation. The remaining sulphuric species in TiO_2_ after its treatment at relatively low temperatures, such as 400 or 450 °C, revealed antimicrobial properties towards *E. coli*.

Conversely, modification of sulphated titania with ammonia solution (NH_4_OH) causes leaching of sulphuric species from its surface and has an adverse effect towards inactivation of *E. coli*. When ammonia species are added to the amorphous titania slurry solution, they are adsorbed on its defect sites, contrary to the modification of already crystalized TiO_2_ with gaseous NH_3_, where only surface coverage with ammonia species occurs. Adsorbed ammonia species can react with sulphate ones and form a new ammonium sulphate compound, which is easily soluble in water. Therefore, TiO_2_ treated with ammonia aqueous solution (NH_4_OH) had the lowest quantity of sulphate species and revealed the lowest activity towards the inactivation of *E. coli*.

In contrast to the modification of TiO_2_ with an aqueous ammonia solution, treatment with gaseous NH_3_ at 400 °C did not lead to the removal of sulphate species, and such a prepared photocatalyst was very active for the inactivation of *E. coli*. Thermal decomposition of the formed ammonium sulphate took place at 450 °C together with desorption of hydroxyl groups. However, sulphated TiO_2_ heat-treated at 450 °C in Ar did not lose its sulphate species. Hydrogenation of TiO_2_ accelerates the growth of anatase crystallites. Hydrogen diffused to the titania interstitial position and formed titanium hydride, which was proven by FTIR analysis. Such modifications of TiO_2_ caused changes in the chemical surface of TiO_2_: the zeta potential shifted towards negative values, the EPR spectra revealed the formation of electron traps near the titania band, and in cases of samples heat-treated at 450 °C, some free electrons in the conductive bands were detected. Such modified TiO_2_ was not suitable for the inactivation of *E. coli*, but the presence of free electrons in TiO_2_ increases the chance of oxygen molecule adsorption on its surface and the formation of superoxide anionic radicals (O_2_^–●^). Therefore, there is a high probability that generated ROS (reactive oxygen species) on TiO_2_ heat-treated at 450 °C in H_2_ had a beneficial impact on the inactivation of *S. epidermidis*.

The formation of Ti^3+^ centres was not observed in TiO_2_ modified with an aqueous ammonia solution (NH_4_OH). Most likely, in this case, some ammonia species were adsorbed on the titania defect sites and formed some nitrogen paramagnetic compounds. These nitrogen species were located above the TiO_2_ valence band and formed an additional Fermi level in the TiO_2_ structure. The changes in the optical properties of such modified TiO_2_ were observed in the UV-Vis/DR spectra recorded. A similar phenomenon was already observed by our group when TiO_2_ was prepared from titanium(IV) isopropoxide in aqueous ammonium solution [28]. However, such TiO_2_ modifications could increase the amount of hole traps, which resulted in an increase in the formation of ∙OH radicals. These hydroxyl radicals could also participate in the oxidation of bacterial cell membranes. However, the lifetime of reactive radicals is short, and therefore, the toxic properties of TiO_2_ towards bacteria increase with an increase of their diffusion to the bacterial surface, which is limited by electrostatic interactions and the size of the TiO_2_ nanoparticles.

Considering that a negative zeta potential at a pH higher than 2 is characteristic of most bacteria, whose outer cell envelope contains a predominance of negatively charged functional groups (originating from peptidoglycan or teichoic teichuronic acid for Gram-positive bacteria or from lipopolysaccharide, phospholipids, and proteins for Gram-negative bacteria), it might seem that the best antibacterial properties should be presented by photocatalysts with a positive zeta potential. This was indeed the case for samples heat-treated at 400 and 450 °C in Ar. However, it should be noted that the pH of the mentioned photocatalyst mixtures were relatively low (<5). In the literature, it is pointed out that *E. coli* grows over a wide range of pHs (pH 4.4 to 9.2); in contrast, *S. epidermidis* grows under acidic pH conditions (ranging from 4 to 6) [29,30]. Based on the results obtained in dark conditions, it can be assumed that such a low pH did not influence bacterial survivability. Furthermore, the bacterial death rate in the presence of positively charged TiO_2_ was higher than the negatively charged ones. Due to the surface of the bacteria being negatively charged, better adherence of bacteria to the photocatalyst took place.

In general, all of the titania samples heated at higher temperatures, such as 450 °C, were more active towards inactivation of *S. epidermidis*, except the one modified with an aqueous ammonium solution. Increasing the temperature of the heat treatment causes an increase of TiO_2_ crystallization and results in better separation of charge carriers in TiO_2_. Modification of TiO_2_ with reducing agents, such as NH_3_ or H_2_ at 450 °C, had an impact on the formation of electron traps near the TiO_2_ band, which further played a role in the formation of some ROS. However, modification of TiO_2_ with an aqueous ammonia solution at 450 °C did not cause any formation of Ti^3+^ centres. This means that some of the electrons formed during TiO_2_ excitation were trapped within the titania band gap, most likely with nitrogen species or formed hydroxyl radicals. It can be concluded that the presence of sulphate species in TiO_2_ and its fast diffusion to the bacterial cell by electrostatic attraction highly determines its killing potential to *E. coli*, but inactivation of *S. epidermidis* is more complex and requires a longer time for oxidation of the membrane cell by formed ROS.

## 5. Conclusions

In the studies presented, sulphated TiO_2_ was modified with reducing agents and tested for inactivation of two bacterial strains: *E. coli* and *S. epidermidis.* It was demonstrated that a sulphur content in TiO_2_ of about 1.5% was beneficial in the context of bacterial inactivation under simulated solar light. The pH of the aqueous TiO_2_ suspension played a significant role in the photocatalytic inactivation of *E. coli*. Sulphated TiO_2_ caused a shift in the pH of the solution towards lower values, around 4.6. At such a low pH, TiO_2_ exhibited a positively charged surface, and its diffusion into the bacterial membrane cell was enhanced by the force of electrostatic attraction. Sulphated TiO_2_ heat-treated at 400 °C in Ar was the most efficient for *E. coli* inactivation, whereas that heated at a higher temperature, i.e., 450 °C, was efficient for *S. epidermidis* inactivation. *E. coli* was completely deactivated after 30 min of solar light irradiation, but *S. epidermidis* was inactivated to only 40%. This difference may be due to the different structures of these two types of bacteria. These studies showed that the presence of sulphate species on TiO_2_ and its acidic surface strongly influenced the survival of *E. coli*, whereas the destruction of *S. epidermidis* cells required harsher oxidation conditions. Therefore, TiO_2_ heat-treated at 450 °C, which had a higher amount of Ti^3+^ centres and potentially a higher ability to generate ROS (reactive oxygen species), was more active for the inactivation of *S. epidermidis*, whereas that obtained at 400 °C with a smaller size of nanoparticles was more active for the inactivation of *E. coli*. Modification of TiO_2_ with reducing agents (NH_3_ or H_2_) did not result in superior abilities of TiO_2_ in hydroxyl radicals’ generation. Although TiO_2_ reduced by NH_3_ or H_2_ exhibited a higher quantity of trapped electrons near the TiO_2_ band with the possibility of enhanced ROS production of other types, its surface was positively charged in an aqueous solution, and this greatly limited diffusion to the bacterial cells.

## Figures and Tables

**Figure 1 materials-17-00066-f001:**
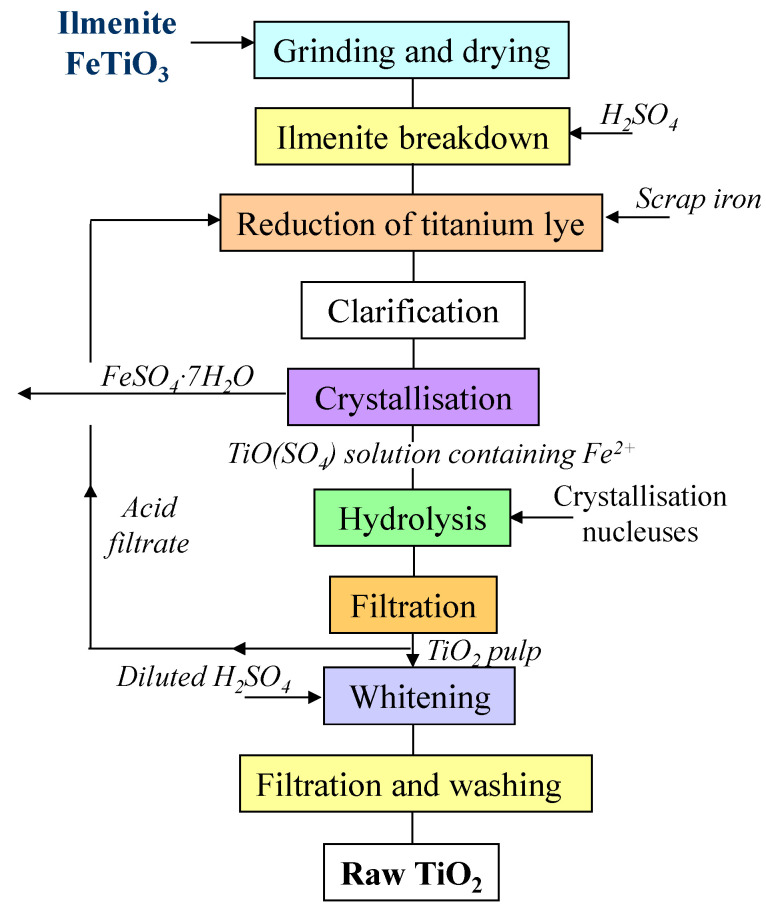
Block diagram of industrial TiO_2_ processing—sulphate process.

**Figure 2 materials-17-00066-f002:**
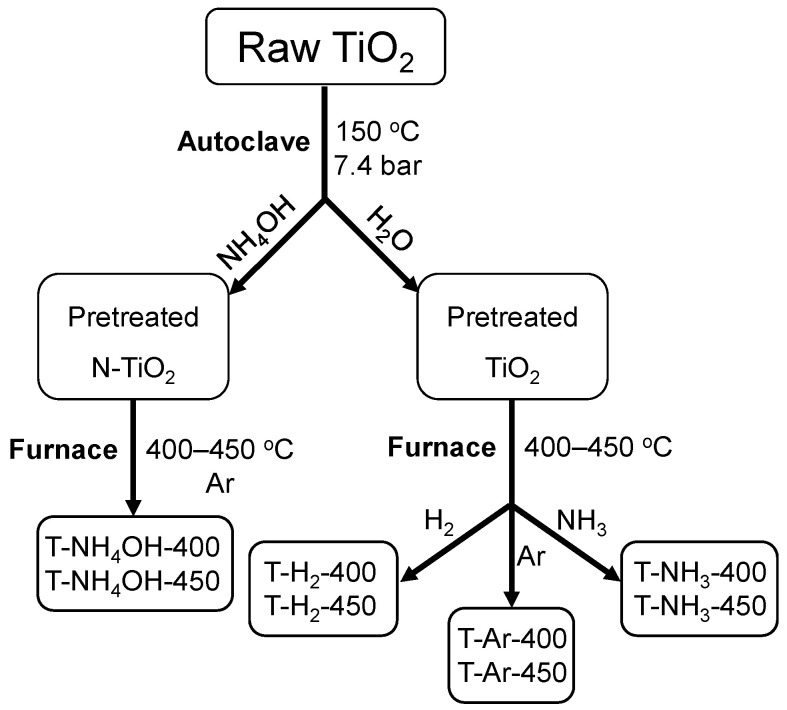
Schematic diagram of the prepared TiO_2_-based photocatalysts.

**Figure 3 materials-17-00066-f003:**
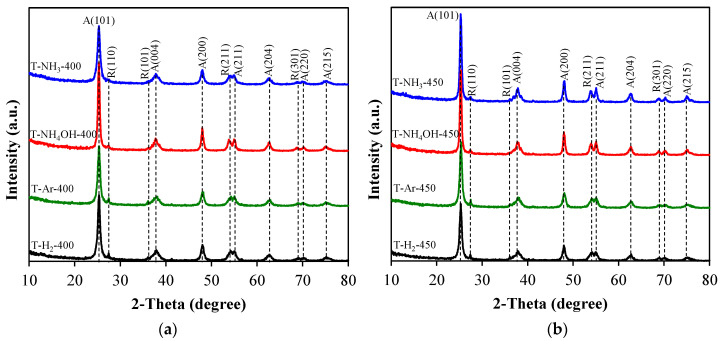
X-Ray diffractograms of TiO_2_-based photocatalysts heat-treated at: (**a**) 400 °C or (**b**) 450 °C. With the symbols A and R, the reflexes from anatase and rutile are indicated, respectively. Miller indexes are given in parentheses.

**Figure 4 materials-17-00066-f004:**
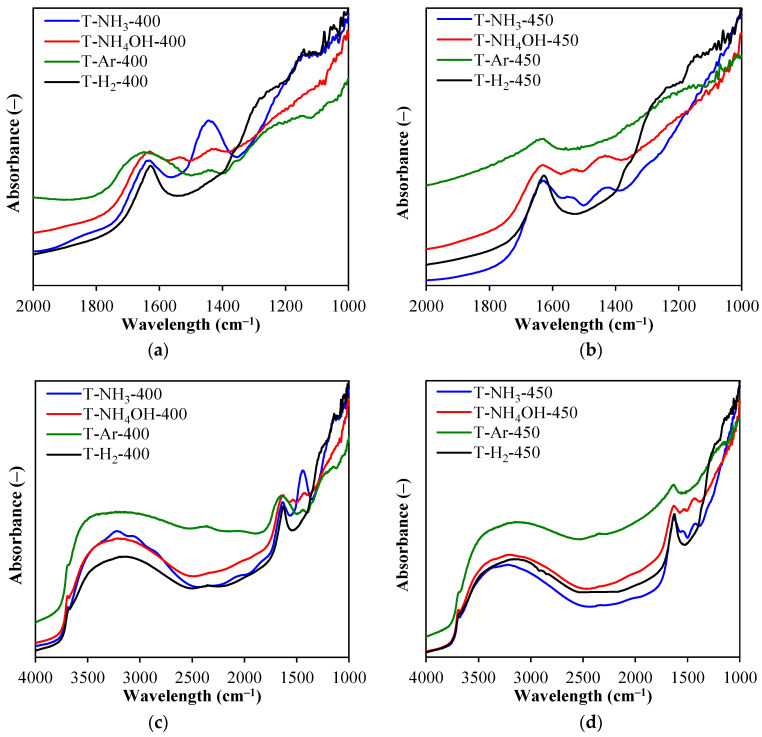
FTIR spectra of the studied samples in the 2000–1000 cm^–1^ range (**a**,**b**) and full range (**c**,**d**).

**Figure 5 materials-17-00066-f005:**
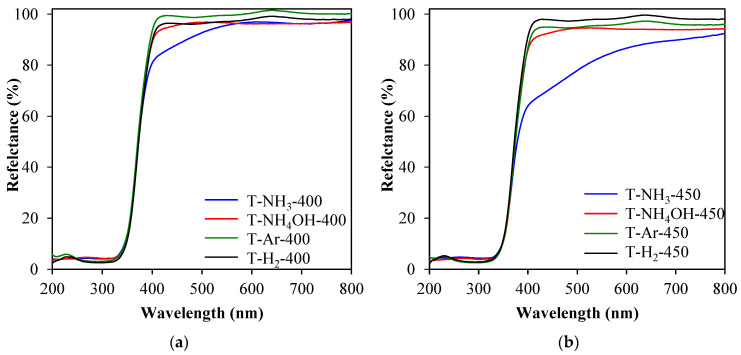
UV-Vis/DR spectra of samples heat-treated at: (**a**) 400 °C or (**b**) 450 °C.

**Figure 6 materials-17-00066-f006:**
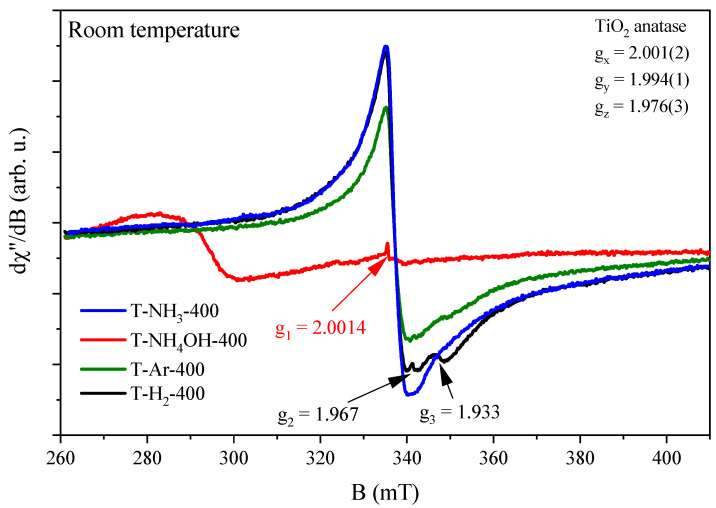
EPR spectra of TiO_2_-based samples heat-treated at 400 °C. The figure shows the positions of the g_1_, g_2_, and g_3_ lines from different magnetic centres and the spin Hamiltonian parameters g_x_, g_y_, and g_z_ for the TiO_2_ in the form of anatase.

**Figure 7 materials-17-00066-f007:**
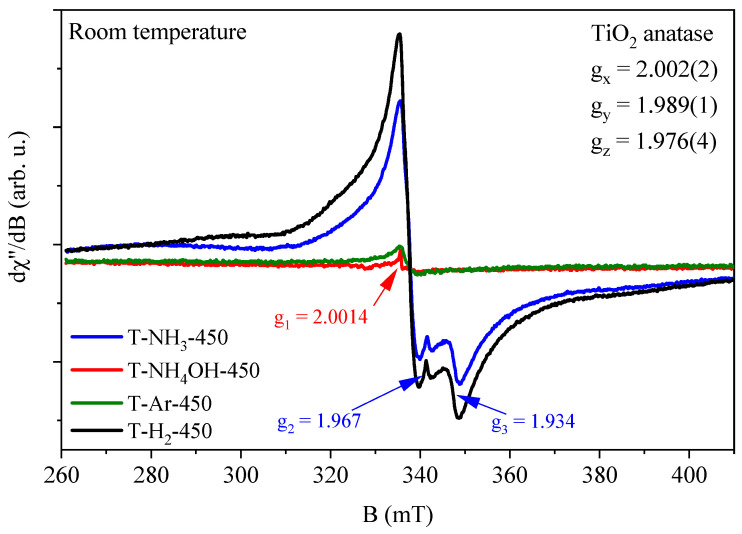
EPR spectra of TiO_2_-based samples heat-treated at 450 °C. The figure shows the positions of the g_1_, g_2_, and g_3_ lines from different magnetic centres and the spin Hamiltonian parameters g_x_, g_y_, and g_z_ for the TiO_2_ in the form of anatase.

**Figure 8 materials-17-00066-f008:**
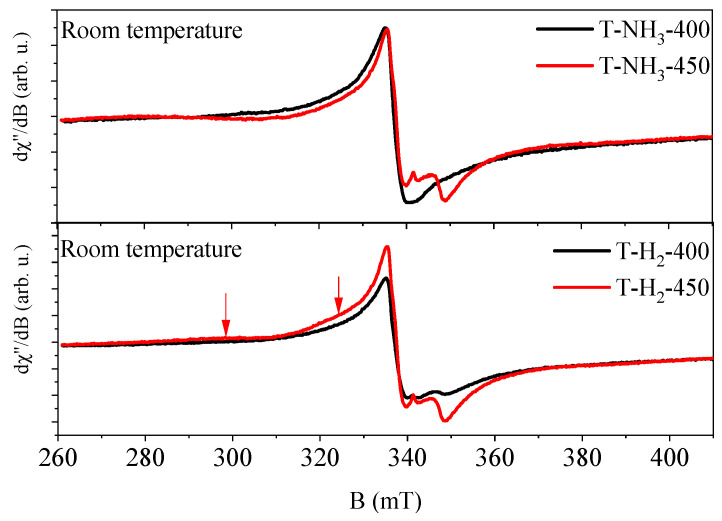
Comparison of the intensity and shape of EPR spectra of TiO_2_-based photocatalysts heat-treated at 400 (black curve) and 450 °C (red curve) in NH_3_ (top panel) and H_2_ (bottom panel). The arrows indicate additional, very broad EPR lines.

**Figure 9 materials-17-00066-f009:**
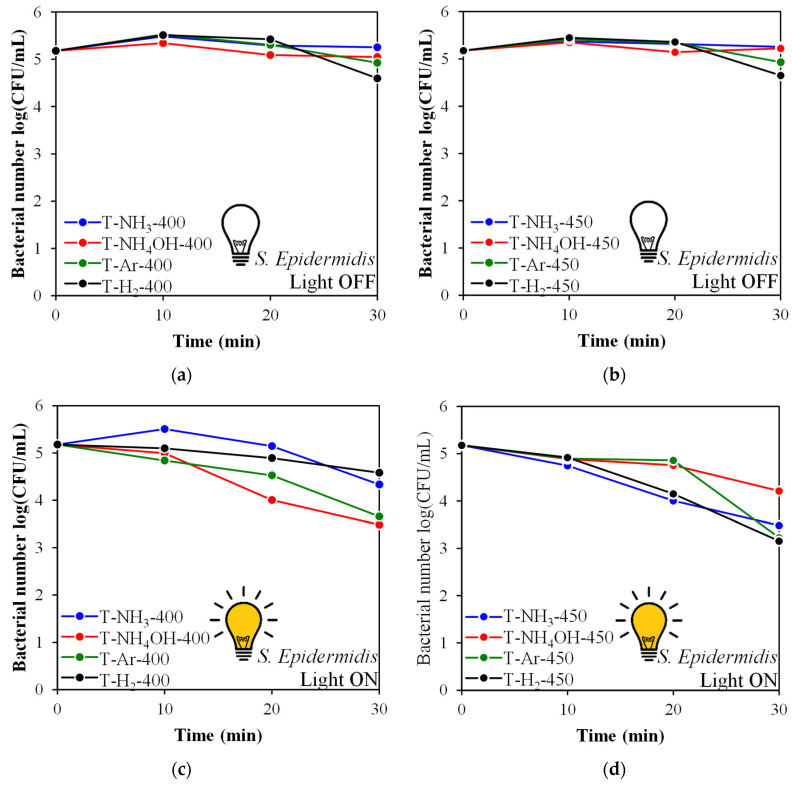
*S. epidermidis* inactivation in the presence of TiO_2_-based photocatalysts: in the dark (**a**,**b**) or irradiated by simulated solar light (**c**,**d**).

**Figure 10 materials-17-00066-f010:**
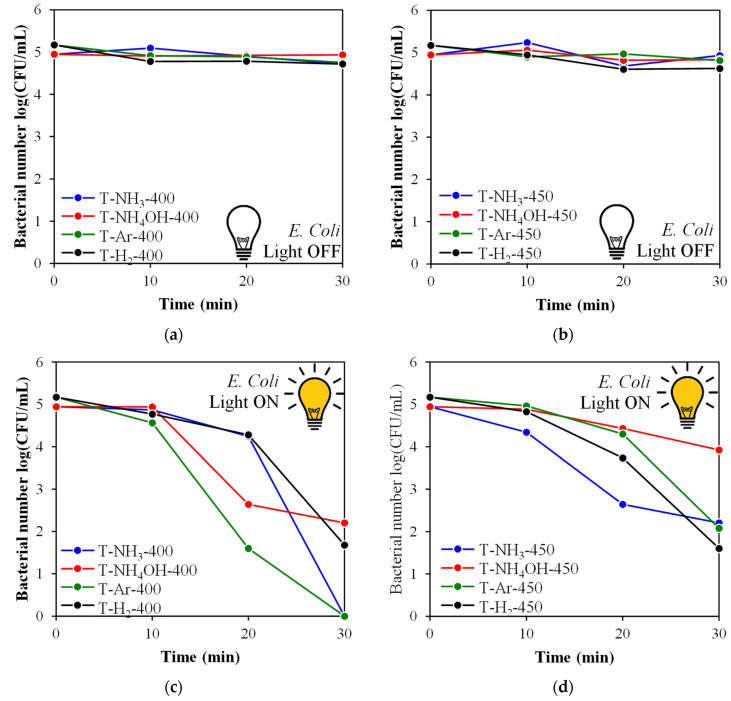
*E. coli* inactivation in the presence of TiO_2_-based photocatalysts: in the dark (**a**,**b**) or irradiated by simulated solar light (**c**,**d**).

**Figure 11 materials-17-00066-f011:**
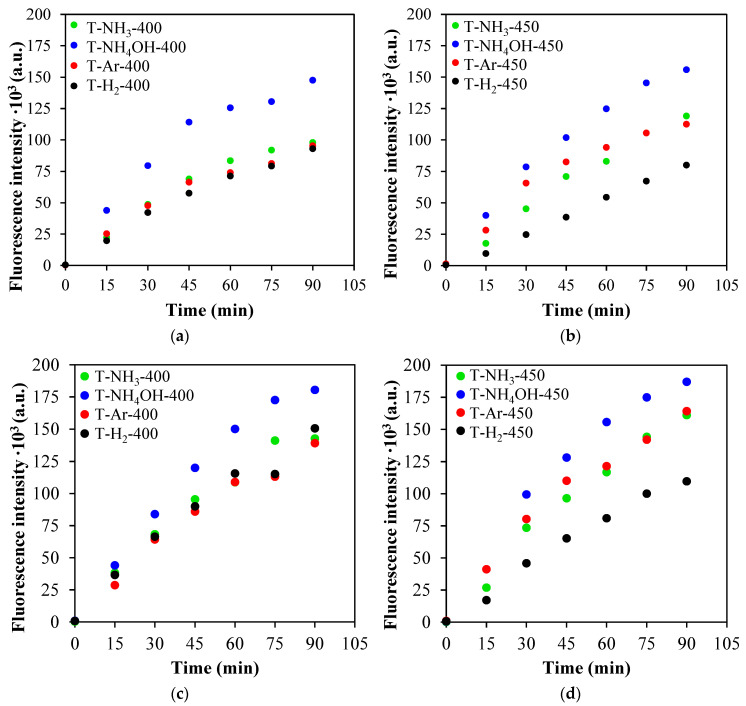
Measurement of the generation of ∙OH radicals by their binding to terephthalic acid under simulated sunlight in: 0.85% NaCl (**a**,**b**) or PBS (**c**,**d**).

**Figure 12 materials-17-00066-f012:**
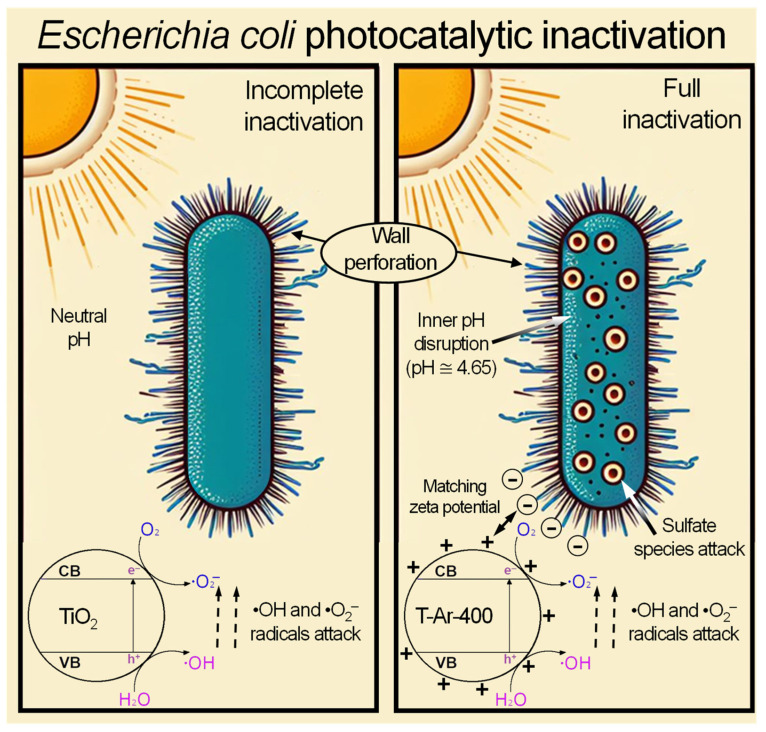
The proposed mechanism for the photocatalytic process of inactivation of the bacteria *E. coli*.

**Table 1 materials-17-00066-t001:** Zeta potential and corresponding pH of prepared TiO_2_ suspensions in 0.85% NaCl.

Sample	pH	Zeta Potential [mV]
T-NH_3_-400	7.10	–6.80
T-NH_4_OH-400	7.13	–6.28
T-Ar-400	4.65	+11.60
T-H_2_-400	7.49	–4.76
T-NH_3_-450	7.11	–9.29
T-NH_4_OH-450	7.01	–7.58
T-Ar-450	4.58	+12.30
T-H_2_-450	7.09	–6.40

**Table 2 materials-17-00066-t002:** XRF sulphur content and crystallites’ size of the prepared TiO_2_-based photocatalysts.

Sample	Sulphur Content	Crystallites’ Size (nm)		Phase Composition (Anatase:Rutile)
		Anatase	Rutile	
T-NH_3_-400	1.19%	12.5	22.9	97:3
T-NH_4_OH-400	0.11%	20.5	19.0	95:5
T-Ar-400	1.40%	15.0	95.2	96:4
T-H_2_-400	1.45%	15.2	70.1	96:4
T-NH_3_-450	950 ppm	20.5	19.7	95:3
T-NH_4_OH-450	0.10%	21.8	37.7	97:3
T-Ar-450	1.46%	15.5	64.5	96:4
T-H_2_-450	0.23%	16.3	64.5	96:4

## Data Availability

Data available upon request.

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
