# Peer review of "Sulphated TiO2 Reduced by Ammonia and Hydrogen as an Excellent Photocatalyst for Bacteria Inactivation"

_materials, 2023, doi:10.3390/ma17010066_

Round 1

Reviewer 1 Report

Comments and Suggestions for Authors

The article entitled “Sulphated TiO2 reduced by ammonia and hydrogen as an excellent photocatalyst for bacteria inactivation” can be accepted for the publication after major revision. Please find my comments below.

1.      Why was modifying TiO2 with reducing agents like ammonia and hydrogen at lower temperatures of 400-450°C considered an attractive approach?

2.      How did the presence of sulfate species on TiO2 shift the pH and impact photocatalytic inactivation of E. coli?

3.      Why was E. coli more readily inactivated than S. epidermidis? What key differences made S. epidermidis more resistant?

4.      How did modification with aqueous ammonia solution versus gaseous ammonia impact the photocatalytic activity towards E. coli inactivation?

5.      How did the crystallinity, sulfur content, and formation of defects in the various TiO2 samples impact their photocatalytic antibacterial activity?

6.      What role did zeta potential play in enabling diffusion of reactive radicals to bacterial cell membranes? 

7.      How did heat treatment temperature impact the generation of ROS and antibacterial activity towards S. epidermidis versus E. coli?

8.      Why was TiO2 modified by hydrogenation less effective for E. coli inactivation despite increased electron traps and potential for enhanced ROS? 

9.      What exactly occurred upon NH3 and H2 modification at 450°C to give free electrons in the conductive band according to the EPR spectra?

10.  Could the hydroxyl radicals generated correlate directly to photocatalytic antibacterial activity, or were other factors more critical in determining efficacy?

11.  What type of kinetic model (e.g. first-order, second-order) describes the photocatalytic inactivation of bacteria like E. coli and S. epidermidis? What parameters are involved?

12.  What mathematical functions can describe the generation rate of different reactive oxygen species (ROS) like hydroxyl radicals or superoxides over irradiation time? What influences the kinetics?

13.  What kind of particle interaction forces or models, such as the Derjaguin-Landau-Verwey-Overbeek (DLVO) theory, determine the surface electrostatic interactions and aggregation of the nanoparticles?

1.      The following studies should be added as references in your manuscript to enhance the quality of your study.

2.      https://doi.org/10.3390/molecules27249032

3.      https://doi.org/10.1016/j.molliq.2020.114696

Comments on the Quality of English Language

Minor editing of the English language required

Reviewer 2 Report

Comments and Suggestions for Authors

Round 2

Reviewer 1 Report

Comments and Suggestions for Authors

Accepted

Reviewer 2 Report

Comments and Suggestions for Authors

Authors made necessary changes in the manuscript.